# Causal Context Connects Counterfactual Fairness to Robust Prediction and Group Fairness

**Jacy Reese Anthis**[1,2,3*], **Victor Veitch**[1]
[1]University of Chicago, [2]University of California, Berkeley, [3]Sentience Institute

## Abstract

Counterfactual fairness requires that a person would have been classified in the same way by an AI or other algorithmic system if they had a different protected class, such as a different race or gender. This is an intuitive standard, as reflected in the U.S. legal system, but its use is limited because counterfactuals cannot be directly observed in real-world data. On the other hand, group fairness metrics (e.g., demographic parity or equalized odds) are less intuitive but more readily observed. In this paper, we use *causal context* to bridge the gaps between counterfactual fairness, robust prediction, and group fairness. First, we motivate counterfactual fairness by showing that there is not necessarily a fundamental trade-off between fairness and accuracy because, under plausible conditions, the counterfactually fair predictor is in fact accuracy-optimal in an unbiased target distribution. Second, we develop a correspondence between the causal graph of the data-generating process and which, if any, group fairness metrics are equivalent to counterfactual fairness. Third, we show that in three common fairness contexts—measurement error, selection on label, and selection on predictors—counterfactual fairness is equivalent to demographic parity, equalized odds, and calibration, respectively. Counterfactual fairness can sometimes be tested by measuring relatively simple group fairness metrics.

## 1 Introduction

The increasing use of artificial intelligence and machine learning in high stakes contexts such as healthcare, hiring, and financial lending has driven widespread interest in algorithmic fairness. A canonical example is risk assessment in the U.S. judicial system, which became well-known after a 2016 investigation into the recidivism prediction tool COMPAS revealed significant racial disparities [2]. There are several metrics one can use to operationalize fairness. These typically refer to a protected class or sensitive label, such as race or gender, and define an equality of prediction rates across the protected class. For example, demographic parity, also known as statistical parity or group parity, requires equal classification rates across all levels of the protected class [7].

However, there is an open challenge of deciding which metrics to enforce in a given context [e.g., 24, 44]. Appropriateness can vary based on different ways to measure false positive and false negative rates and the costs of such errors [19] as well as the potential trade-offs between fairness and accuracy [11, 12, 21, 55]. Moreover, there are well-known technical results showing that it is impossible to achieve the different fairness metrics simultaneously, even to an $\epsilon$-approximation [10, 30], which suggest that a practitioner's context-specific perspective may be necessary to achieve satisfactory outcomes [4].

A different paradigm, counterfactual fairness, requires that a prediction would have been the same if the person had a different protected class [31]. This matches common intuitions and legal standards

---

*Corresponding author: anthis@uchicago.edu

37th Conference on Neural Information Processing Systems (NeurIPS 2023).

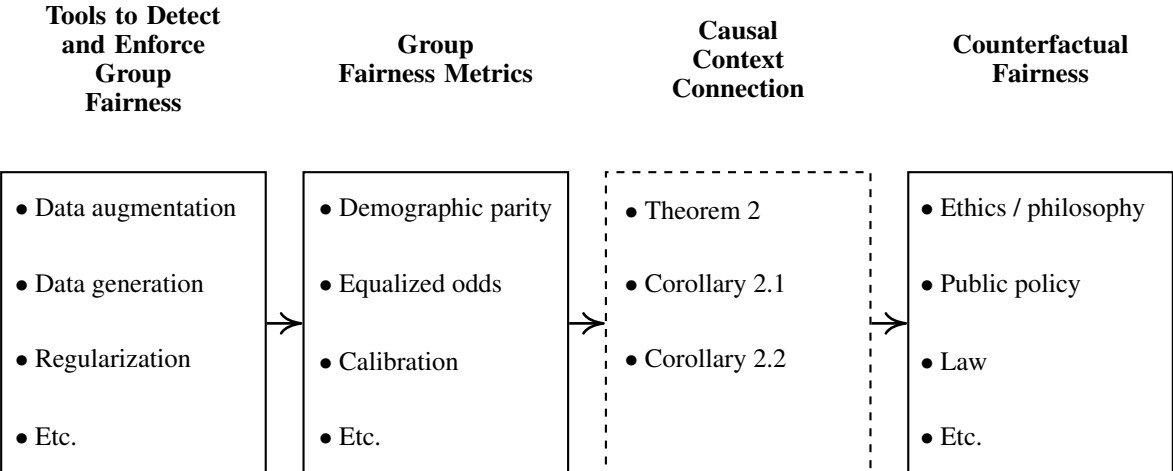

Figure 1: A pipeline to detect and enforce counterfactual fairness. This is facilitated by Theorem 2, a correspondence between group fairness metrics and counterfactual fairness, such that tools to detect and enforce group fairness can be applied to counterfactual fairness.

[25]. For example, in *McCleskey v. Kemp* (1987), the Supreme Court found "disproportionate impact" as shown by group disparity to be insufficient evidence of discrimination and required evidence that "racial considerations played a part." The Court found that there was no merit to the argument that the Georgia legislature had violated the Equal Protection Clause via its use of capital punishment on the grounds that it had not been shown that this was [sic] "*because* of an anticipated racially discriminatory effect." While judges do not formally specify causal models, their language usually evokes such counterfactuals; see, for example, the test of "but-for causation" in *Bostock v. Clayton County* (2020).

We take a new perspective on counterfactual fairness by leveraging *causal context*. First, we provide a novel motivation for counterfactual fairness from the perspective of robust prediction by showing that, with certain causal structures of the data-generating process, counterfactually fair predictors are accuracy-optimal in an unbiased target distribution (Theorem 1). The contexts we consider ihave two important features: the faithfulness of the causal structure, in the sense that no causal effects happen to be precisely counterbalanced, which could lead to a coincidental achievement of group fairness metrics, and that the association between the label $Y$ and the protected class $A$ is "purely spurious," in the sense that intervening on $A$ does not affect $Y$ or vice versa.

Second, we address the fundamental challenge in enforcing counterfactual fairness, which is that, by definition, we never directly observe counterfactual information in the real world. To deal with this, we show that the causal context connects counterfactual fairness to observational group fairness metrics (Theorem 2 and Corollary 2.1), and this correspondence can be used to apply tools from group fairness frameworks to the ideal of achieving counterfactual fairness (Figure 1). For example, the fairness literature has developed techniques to achieve group fairness through augmenting the input data [13, 27, 42], data generation [1], or regularization [45]. With this pipeline, these tools can be applied to enforce counterfactual fairness by achieving the specific group fairness metric that corresponds to counterfactual fairness in the given context. In particular, we show that in each fairness context shown in Figure 2, counterfactual fairness is equivalent to a particular metric (Corollary 2.2). This correspondence can be used to apply tools built for group fairness to the goal of counterfactual fairness or vice versa.

Finally, we conduct brief experiments in a semi-synthetic setting with the Adult income dataset [3] to confirm that a counterfactually fair predictor under these conditions achieves out-of-distribution accuracy and the corresponding group fairness metric. To do this, we develop a novel counterfactually fair predictor that is a weighted average of naive predictors, each under the assumption that the observation is in each protected class (Theorem 3).

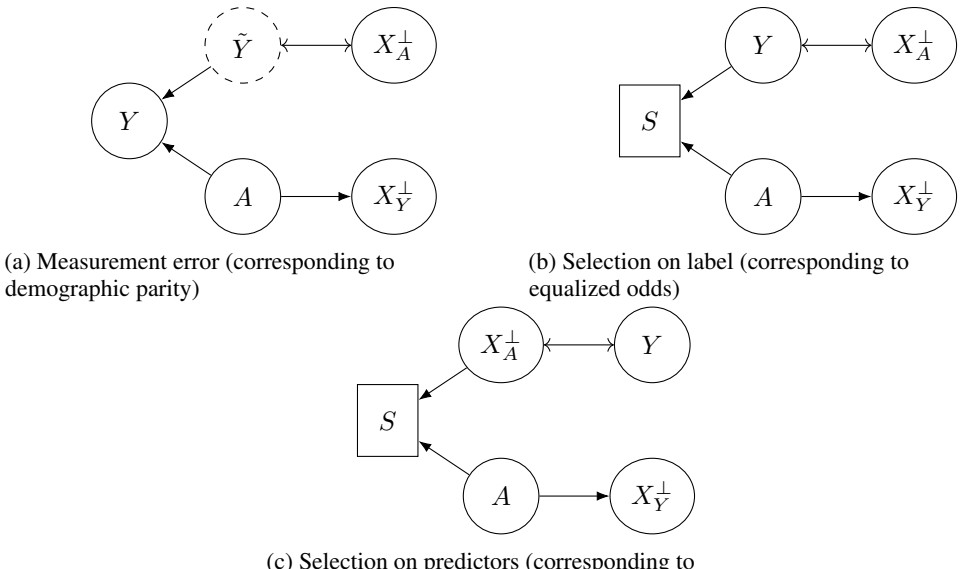

(a) Measurement error (corresponding to demographic parity)

(b) Selection on label (corresponding to equalized odds)

(c) Selection on predictors (corresponding to calibration)

Figure 2: DAGs for three causal contexts in which a counterfactually fair predictor is equivalent to a particular group fairness metric. Measurement error is represented with the unobserved true label $\tilde{Y}$ in a dashed circle, which is a parent of the observed label $Y$, and selection is represented with the selection status $S$ in a rectangle, indicating an observed variable that is conditioned on in the data-generating process, which induces an association between its parents. Bidirectional arrows indicate that either variable could affect the other.

In summary, we make three main contributions:

1. We provide a novel motivation for counterfactual fairness from the perspective of robust prediction by showing that, in certain causal contexts, the counterfactually fair predictor is accuracy-optimal in an unbiased target distribution (Theorem 1).

2. We provide criteria for whether a causal context implies equivalence between counterfactual fairness and each of the three primary group fairness metrics (Theorem 2) as well as seven derivative metrics (Corollary 2.1).

3. We provide three causal contexts, as shown in Figure 2, in which counterfactual fairness is equivalent to a particular group fairness metric: measurement error with demographic parity, selection on label with equalized odds, and selection on predictors with calibration (Corollary 2.2).

For a running example, consider the case of recidivism prediction with measurement error, as shown in Figure 2a, where $X$ is the set of predictors of recidivism, such as past convictions; $\tilde{Y}$ is committing a crime; $Y$ is committing a crime that has been reported and prosecuted; and $A$ is race, which in this example affects the likelihood crime is reported and prosecuted but not the likelihood of crime itself. In this case, we show that a predictor is counterfactually fair if and only if it achieves demographic parity.

## 2 Related work

Previous work has developed a number of fairness metrics, particularly demographic parity [7], equalized odds [23], and calibration [19]. Verma and Rubin [48] details 15 group fairness metrics—also known as observational or distributional fairness metrics—including these three, and Makhlouf, Zhioua, and Palamidessi [34] details 19 causal fairness metrics, including counterfactual fairness [31]. Many notions of fairness can be viewed as specific instances of the general goal of invariant representation [18, 53, 55, 56]. Much contemporary work in algorithmic fairness develops ways to better enforce certain fairness metrics, such as through reprogramming a pretrained model [54],

contrastive learning when most data lacks labels of the protected class [8], and differentiating between critical and non-critical features for selective removal [16].

Several papers have criticized the group fairness metrics and enriched them correspondingly. For example, one can achieve demographic parity by arbitrarily classifying a subset of individuals in a protected class, regardless of the other characteristics of those individuals. This is particularly important for subgroup fairness, in which a predictor can achieve the fairness metric for one category within the protected class but fail to achieve it when subgroups are considered across different protected classes, such as people who have a certain race and gender. This is particularly important given the intersectional nature of social disparities [14], and metrics have been modified to include possible subgroups [29].

Well-known impossibility theorems show that fairness metrics such as the three listed above cannot be simultaneously enforced outside of unusual situations such as a perfectly accurate classifier, a random classifier, or a population with equal rates across the protected class, and this is true even to an $\epsilon$-approximation [10, 30, 40]. Moreover, insofar as fairness and classification accuracy diverge, there appears to be a trade-off between the two objectives. Two well-known papers in the literature, Corbett-Davies et al. [12] and Corbett-Davies and Goel [11], detail the cost of fairness and argue for the prioritization of classification accuracy. More recent work on this trade-off includes a geometric characterization and Pareto optimal frontier for invariance goals such as fairness [55], the cost of causal fairness in terms of Pareto dominance in classification accuracy and diversity [37], the potential for increases in fairness via data reweighting in some cases without a cost to classification accuracy [32], and showing optimal fairness and accuracy in an ideal distribution given mismatched distributional data [17].

Our project uses the framework of causality to enrich fairness metric selection. Recent work has used structural causal models to formalize and solve a wide range of causal problems in machine learning, such as disentangled representations [41, 50], out-of-distribution generalization [41, 49, 50], and the identification and removal of spurious correlations [47, 50]. While Veitch et al. [47] focuses on the correspondence between counterfactual invariance and domain generalization, Remark 3.3 notes that counterfactual fairness as an instance of counterfactual invariance can imply either demographic parity or equalized odds in the two graphs they analyze, respectively, and Makar and D'Amour [33] draw the correspondence between risk invariance and equalized odds under a certain graph. The present work can be viewed as a generalization of these results to correspondence between counterfactual fairness, which is equivalent to risk invariance when the association is "purely spurious," and each of the group fairness metrics.

## 3 Preliminaries and Examples

For simplicity of exposition, we focus on binary classification with an input dataset $X \in \mathcal{X}$ and a binary label $Y \in \mathcal{Y} = \{0, 1\}$, and in our data we may have ground-truth unobserved labels $\tilde{Y} \in \{0, 1\}$ that do not always match the potentially noisy observed labels $Y$. The goal is to estimate a prediction function $f : \mathcal{X} \mapsto \mathcal{Y}$, producing estimated labels $f(X) \in \{0, 1\}$, that minimizes $\mathbb{E}[\ell(f(X), Y)]$ for some loss function $\ell : \mathcal{Y} \times \mathcal{Y} \mapsto \mathbb{R}$. There are a number of popular fairness metrics in the literature that can be formulated as conditional probabilities for some protected class $A \in \{0, 1\}$, or equivalently, as independence of random variables, and we can define a notion of counterfactual fairness.

**Definition 1.** (Demographic parity). A predictor $f(X)$ achieves demographic parity if and only if:

$$\mathbb{P}(f(X) = 1 \mid A = 0) = \mathbb{P}(f(X) = 1 \mid A = 1) \iff f(X) \perp A$$

**Definition 2.** (Equalized odds). A predictor $f(X)$ achieves equalized odds if and only if:

$$\mathbb{P}(f(X) = 1 \mid A = 0, Y = y) = \mathbb{P}(f(X) = 1 \mid A = 1, Y = y) \iff f(X) \perp A \mid Y$$

**Definition 3.** (Calibration). A predictor $f(X)$ achieves calibration if and only if:

$$\mathbb{P}(Y = 1 \mid A = 0, f(X) = 1) = \mathbb{P}(Y = 1 \mid A = 1, f(X) = 1) \iff Y \perp A \mid f(X)$$

**Definition 4.** (Counterfactual fairness). A predictor $f(X)$ achieves counterfactual fairness if and only if for any $a, a' \in A$:

$$f(X(a)) = f(X(a'))$$

In this case, $x(0)$ and $x(1)$ are the potential outcomes. That is, for an individual associated with data $x \in X$ with protected class $a \in A$, their counterfactual is $x(a')$ where $a' \in A$ and $a \neq a'$. The counterfactual observation is what would obtain if the individual had a different protected class. We represent causal structures as directed acyclic graphs (DAGs) in which nodes are variables and directed arrows indicates the direction of causal effect. Nodes in a solid circle are observed variables that may or may not be included in a predictive model. A dashed circle indicates an unobserved variable, such as in the case of measurement error in which the true label $\tilde{Y}$ is a parent of the observed label $Y$. A rectangle indicates an observed variable that is conditioned on in the data-generating process, typically denoted by $S$ for a selection effect, which induces an association between its parents. For clarity, we decompose $X$ into $X_A^{\perp}$, the component that is not causally affected by $A$, and $X_Y^{\perp}$, the component that does not causally affect $Y$ (in a causal-direction graph, i.e., $X$ affects $Y$) or is not causally affected by $Y$ (in an anti-causal graph, i.e., $Y$ affects $X$). In general, there could be a third component that is causally connected to both $A$ and $Y$ (i.e., a non-spurious interaction effect). This third component is excluded through the assumption that the association between $Y$ and $A$ in the training distribution is *purely spurious* [47].

**Definition 5.** (Purely spurious). We say the association between $Y$ and $A$ is purely spurious if $Y \perp X \mid X_A^{\perp}, A$.

In other words, if we condition on $A$, then the only information about $Y$ is in $X_A^{\perp}$ (i.e., the component of the input that is not causally affected by the protected class). We also restrict ourselves to cases in which $A$ is exogenous to the causal structure (i.e., there is no confounder of $A$ and $X$ or of $A$ and $Y$), which seems reasonable in the case of fairness because the protected class is typically not caused by other variables in the specified model.

Despite the intuitive appeal and legal precedent for counterfactual fairness, the determination of counterfactuals is fundamentally contentious because, by definition, we do not observe counterfactual worlds, and we cannot run scientific experiments to test what would have happened in a different world history. Some counterfactuals are relatively clear, but counterfactuals in contexts of protected classes such as race, gender, and disability can be ambiguous. Consider Bob, a hypothetical Black criminal defendant being assessed for recidivism who is determined to be high-risk by a human judge or a machine algorithm. The assessment involves extensive information about his life: where he has lived, how he has been employed, what crimes he has been arrested and convicted of before, and so on. What is the counterfactual in which Bob is White? Is it the world in which Bob was born to White parents, the one in which Bob's parents were themselves born to White parents, or another change further back? Would Bob still have the same educational and economic circumstances? Would those White parents have raised Bob in the same majority-Black neighborhood he grew up in or in a majority-White neighborhood? Questions of counterfactual ambiguity do not have established answers in the fairness literature, either in philosophy [35], epidemiology [46], or the nascent machine learning literature [28], and we do not attempt to resolve them in the present work.

Additionally, defining counterfactual fairness in a given context requires some identification of the causal structure of the data generating process. Fortunately, mitigating this challenge is more technically tractable than mitigating ambiguity, as there is an extensive literature on how we can learn about the causal structure, known as as causal inference [15] or causal discovery when we have minimal prior knowledge or assumptions of which parent-child relationships do and do not obtain [43]. The literature also contains a number of methods for assessing counterfactual fairness given a partial or complete causal graph [9, 20, 51, 52, 57]. Again, we do not attempt to resolve debates about appropriate causal inference strategies in the present work, but merely to provide a conceptual tool for those researchers and practitioners who are comfortable staking a claim of some partial or complete causal structure of the context at hand.

To ground our technical results, we briefly sketch three fairness contexts that match those in Figure 2 and will be the basis of Corollary 2.2. First, measurement error [6] is a well-known source of fairness issues as developed in the "measurement error models" of Jacobs and Wallach [26]. Suppose that COMPAS is assessing the risk of recidivism, but they do not have perfect knowledge of who has committed crimes because not all crimes are reported and prosecuted. Thus, $X$ causes $\tilde{Y}$, the actual committing of a crime, which is one cause of $Y$, the imperfect labels. Also suppose that the protected class $A$ affects whether the crime is reported and prosecuted, such as through police bias, but does not affect the actual committing of a crime. $A$ also affects $X$, other data used for the recidivism

prediction. This is represented by the causal DAG in Figure 2a, which connects counterfactual fairness to demographic parity.

Second, a protected class can affect whether individuals are selected into the dataset. For example, drugs may be prescribed on the basis of whether the individual's lab work $X$ indicates the presence of disease $Y$. People with the disease are presumably more likely to have lab work done, and the socioeconomic status $A$ of the person (or of the hospital where their lab work is done) may make them more likely to have lab work done. We represent this with a selection variable $S$ that we condition on by only using the lab work data that is available. This collider induces an association between $A$ and $Y$. This is represented by the causal DAG in Figure 2b, which connects counterfactual fairness to equalized odds.

Third, individuals may be selected into the dataset based on predictors, $X$, which cause the label $Y$. For example, loans may be given out with a prediction of loan repayment $Y$ based on demographic and financial records $X$. There may be data only for people who choose to work with a certain bank based on financial records and a protected class $A$. This is represented by the causal DAG in Figure 2c, which connects counterfactual fairness to calibration.

## 4 Counterfactual fairness and robust prediction

Characterizations of the trade-off between prediction accuracy and fairness treat them as two competing goals [e.g., 11, 12, 21, 55]. That view assumes the task is to find an optimal model $f^*(X)$ that minimizes risk (i.e., expected loss) in the training distribution $X, Y, A \sim P$:

$$f^*(X) = \underset{f}{\operatorname{argmin}} \, \mathbb{E}_P[\ell(f(X), Y)]$$

However, the discrepancy between levels of the protected class in the training data may itself be due to biases in the dataset, such as measurement error, selection on label, and selection on predictors. As such, the practical interest may not be risk minimization in the training distribution, but out-of-distribution (OOD) generalization from the training distribution to an unbiased target distribution where these effects are not present.

Whether the counterfactually fair empirical risk minimizer also minimizes risk in the target distribution depends on how the distributions differ. Because a counterfactually fair predictor does not depend on the protected class, it minimizes risk if the protected class in the target distribution contains no information about the corresponding labels (i.e., if protected class and label are uncorrelated). If the protected class and label are correlated in the target distribution, then risk depends on whether the counterfactually fair predictor learns all the information about the label contained in the protected class. If so, then its prediction has no reason to vary in the counterfactual scenario. Theorem 1 motivates counterfactual fairness by stating that, for a distribution with bias due to selection on label and equal marginal label distributions or due to selection on predictors, the counterfactually fair predictor is accuracy-optimal in the unbiased target distribution.

**Theorem 1.** *Let $\mathcal{F}^{CF}$ be the set of all counterfactually fair predictors. Let $\ell$ be a proper scoring rule (e.g., square error, cross entropy loss). Let the counterfactually fair predictor that minimizes risk on the training distribution $X, Y, A \sim P$ be:*

$$f^*(X) := \underset{f \in \mathcal{F}^{CF}}{\operatorname{argmin}} \, \mathbb{E}_P[\ell(f(X), Y)]$$

*Then, $f^*$ also minimizes risk on the target distribution $X, Y, A \sim Q$ with no selection effects, i.e.,*

$$f^*(X) = \underset{f}{\operatorname{argmin}} \, \mathbb{E}_Q[\ell(f(X), Y)]$$

*if either of the following conditions hold:*

1. *The association between $Y$ and $A$ is due to selection on label and the marginal distribution of the label $Y$ is the same in each distribution, i.e., $P(Y) = Q(Y)$.*
2. *The association between $Y$ and $A$ is due to selection on predictors.*

In the context of measurement error, there are not straightforward conditions for the risk minimization of the counterfactually fair predictor because the training dataset contains $Y$, observed noisy labels,

and not $\tilde{Y}$, the true unobserved labels. Thus any risk minimization (including in the training distribution) depends on the causal process that generates $Y$ from $\tilde{Y}$.

# 5 Counterfactual fairness and group fairness

Causal structures imply conditional independencies. For example, if the only causal relationships are that $X$ causes $Y$ and $Y$ causes $Z$, then we know that $X \perp Z \mid Y$. On a directed acyclic graph (DAG), following Pearl and Dechter [38], we say that a variable $Y$ **dependence-separates** or **d-separates** $X$ and $Z$ if $X$ and $Z$ are connected via an unblocked path (i.e., no unconditioned **collider** in which two arrows point directly to the same variable) but are no longer connected after removing all arrows that directly connect to $Y$ [22]. So in this example, $Y$ d-separates $X$ and $Z$, which is equivalent to the conditional independence statement. A selection variable $S$ in a rectangle indicates an observed variable that is conditioned on in the data-generating process, which induces an association between its parents and thereby does not block the path as an unconditioned collider would. For Theorem 2, Corollary 2.1, and Corollary 2.2, we make the usual assumption of faithfulness of the causal graph, meaning that the only conditional independencies are those implied by d-separation, rather than any causal effects that happen to be precisely counterbalanced. Specifically, we assume faithfulness between the protected class $A$, the label $Y$, and the component of $X$ on which the predictor $f(X)$ is based. If the component is not already its own node in the causal graph, then faithfulness would apply if the component were isolated into its own node or nodes.

For a predictor, these implied conditional independencies can be group fairness metrics. We can restate conditional independencies containing $X_A^{\perp}$ as containing $f(X)$ because, if $f(X)$ is a counterfactually fair predictor, it only depends on $X_A^{\perp}$, the component that is not causally affected by $A$. In Theorem 2, we provide the correspondence between counterfactual fairness and the three most common metrics: demographic parity, equalized odds, and calibration.

**Theorem 2.** *Let the causal structure be a causal DAG with $X_Y^{\perp}$, $X_A^{\perp}$, $Y$, and $A$, such as in Figure 2. Assume faithfulness between $A$, $Y$, and $f(X)$. Then:*

1. *Counterfactual fairness is equivalent to demographic parity if and only if there is no unblocked path between $X_A^{\perp}$ and $A$.*

2. *Counterfactual fairness is equivalent to equalized odds if and only if all paths between $X_A^{\perp}$ and $A$, if any, are either blocked by a variable other than $Y$ or unblocked and contain $Y$.*

3. *Counterfactual fairness is equivalent to calibration if and only if all paths between $Y$ and $A$, if any, are either blocked by a variable other than $X_A^{\perp}$ or unblocked and contain $X_A^{\perp}$.*

Similar statements can be made for any group fairness metric. For example, the notion of false negative error rate balance, also known as equal opportunity [23], is identical to equalized odds but only considers individuals who have a positive label ($Y = 1$). The case for this metric is based on false negatives being a particular moral or legal concern, motivated by principles such as "innocent until proven guilty," in which a false negative represents an innocent person ($Y = 1$) who is found guilty ($f(X) = 0$). False negative error rate balance is assessed in the same way as equalized odds but only with observations that have a positive true label, which may be consequential if different causal structures are believed to obtain for different groups.

Table 1 shows the ten group fairness metrics presented in Verma and Rubin [48] that can be expressed as conditional independencies. Theorem 2 specified the correspondence between three of these (demographic parity, equalized odds, and calibration), and we extend to the other seven in Corollary 2.1.

We have so far restricted ourselves to a binary classifier $f(X) \in \{0, 1\}$. Here, we denote this as a decision $f(x) = d \in D = \{0, 1\}$ and also consider probabilistic classifiers that produce a score $s$ that can take any probability from 0 to 1, i.e., $f(x) = s \in S = [0, 1] = \mathbb{P}(Y = 1)$. The metrics of balance for positive class, balance for negative class, and score calibration are defined by Verma and Rubin [48] in terms of score. Verma and Rubin [48] refer to calibration for binary classification (Definition 3) as "conditional use accuracy equality." To differentiate these, we henceforth refer to the binary classification metric as "binary calibration" and the probabilistic classification metric as "score calibration."

Table 1: Group fairness metrics from Verma and Rubin [48].

| Name | Probability Definition | Independence Definition |
|------|------------------------|-------------------------|
| Demographic Parity | $\mathbb{P}(D = 1 \mid A = 0) = \mathbb{P}(D = 1 \mid A = 1)$ | $D \perp A$ |
| Conditional Demographic Parity | $\mathbb{P}(D = 1 \mid A = 0, L = l) = \mathbb{P}(D = 1 \mid A = 1, L = l)$ | $D \perp A \mid L = l$ |
| Equalized Odds | $\mathbb{P}(D = 1 \mid A = 0, Y = y) = \mathbb{P}(D = 1 \mid A = 1, Y = y)$ | $D \perp A \mid Y$ |
| False Positive Error Rate Balance | $\mathbb{P}(D = 1 \mid A = 0, Y = 0) = \mathbb{P}(D = 1 \mid A = 1, Y = 0)$ | $D \perp A \mid Y = 0$ |
| False Negative Error Rate Balance | $\mathbb{P}(D = 0 \mid A = 0, Y = 1) = \mathbb{P}(D = 0 \mid A = 1, Y = 1)$ | $D \perp A \mid Y = 1$ |
| Balance for Negative Class | $\mathbb{E}[S \mid A = 0, Y = 0] = \mathbb{E}[S \mid A = 1, Y = 0]$ | $S \perp A \mid Y = 0$ |
| Balance for Positive Class | $\mathbb{E}[S \mid A = 0, Y = 1] = \mathbb{E}[S \mid A = 1, Y = 1]$ | $S \perp A \mid Y = 1$ |
| Conditional Use Accuracy Equality (i.e., Calibration) | $\mathbb{P}(Y = y \mid A = 0, D = d) = \mathbb{P}(Y = y \mid A = 1, D = d)$ | $Y \perp A \mid D$ |
| Predictive Parity | $\mathbb{P}(Y = 1 \mid A = 0, D = 1) = \mathbb{P}(Y = 1 \mid A = 1, D = 1)$ | $Y \perp A \mid D = 1$ |
| Score Calibration | $\mathbb{P}(Y = 1 \mid A = 0, S = s) = \mathbb{P}(Y = 1 \mid A = 1, S = s)$ | $Y \perp A \mid S$ |

**Corollary 2.1.** *Let the causal structure be a causal DAG with $X_Y^{\perp}$, $X_A^{\perp}$, $Y$, and $A$, such as in Figure 2. Assume faithfulness between $A$, $Y$, and $f(X)$. Then:*

1. *For a binary classifier, counterfactual fairness is equivalent to **conditional demographic parity** if and only if, when a set of legitimate factors $L$ is held constant at level $l$, there is no unblocked path between $X_A^{\perp}$ and $A$.*

2. *For a binary classifier, counterfactual fairness is equivalent to **false positive error rate balance** if and only if, for the subset of the population with negative label (i.e., $Y = 0$), there is no path between $X_A^{\perp}$ and $A$, a path blocked by a variable other than $Y$, or an unblocked path that contains $Y$.*

3. *For a binary classifier, counterfactual fairness is equivalent to **false negative error rate balance** if and only if, for the subset of the population with positive label (i.e., $Y = 1$), there is no path between $X_A^{\perp}$ and $A$, a path blocked by a variable other than $Y$, or an unblocked path that contains $Y$.*

4. *For a probabilistic classifier, counterfactual fairness is equivalent to **balance for negative class** if and only if, for the subset of the population with negative label (i.e., $Y = 0$), there is no path between $X_A^{\perp}$ and $A$, a path blocked by a variable other than $Y$, or an unblocked path that contains $Y$.*

5. *For a probabilistic classifier, counterfactual fairness is equivalent to **balance for positive class** if and only if, for the subset of the population with positive label (i.e., $Y = 1$), there is no path between $X_A^{\perp}$ and $A$, a path blocked by a variable other than $Y$, or an unblocked path that contains $Y$.*

6. *For a probabilistic classifier, counterfactual fairness is equivalent to **predictive parity** if and only if, for the subset of the population with positive label (i.e., $D = 1$), there is no path between $Y$ and $A$, a path blocked by a variable other than $X_A^{\perp}$, or an unblocked path that contains $X_A^{\perp}$.*

7. *For a probabilistic classifier, counterfactual fairness is equivalent to **score calibration** if and only if there is no path between $Y$ and $A$, a path blocked by a variable other than $X_A^{\perp}$, or an unblocked path that contains $X_A^{\perp}$.*

Table 2: Experimental results: Robust prediction

| | Source Accuracy | | | Target Accuracy | | | |
|---|---|---|---|---|---|---|---|
| | **Naive** | **FTU** | **CF** | **Naive** | **FTU** | **CF** | **Target Trained** |
| Measurement Error | 0.8283 | 0.7956 | 0.7695 | 0.8031 | 0.8114 | 0.8204 | 0.8674 |
| Selection on Label | 0.8771 | 0.8686 | 0.8610 | 0.8566 | 0.8652 | 0.8663 | 0.8655 |
| Selection on Predictors | 0.8659 | 0.656 | 0.8658 | 0.8700 | 0.8698 | 0.8699 | 0.8680 |

This general specification can be unwieldy, so to illustrate this, we provide three real-world examples shown in Figure 2: measurement error, selection on label (i.e., "post-treatment bias" or "selection on outcome" if the label $Y$ is causally affected by $X$), and selection on predictors. These imply demographic parity, equalized odds, and calibration, respectively.

**Corollary 2.2.** *Assume faithfulness between $A$, $Y$, and $f(X)$.*

1. *Under the graph with measurement error as shown in Figure 2a, a predictor achieves counterfactual fairness if and only if it achieves demographic parity.*

2. *Under the graph with selection on label as shown in Figure 2b, a predictor achieves counterfactual fairness if and only if it achieves equalized odds.*

3. *Under the graph with selection on predictors as shown in Figure 2c, a predictor achieves counterfactual fairness if and only if it achieves calibration.*

## 6 Experiments

We conducted brief experiments in a semi-synthetic setting to show that a counterfactually fair predictor achieves robust prediction and group fairness. We used the Adult income dataset [3] with a simulated protected class $A$, balanced with $P(A = 0) = P(A = 1) = 0.5$. For observations with $A = 1$, we manipulated the input data to simulate a causal effect of $A$ on $X$: $P(\texttt{race} = \texttt{Other}) = 0.8$. With this as the target distribution, we produced three biased datasets that result from each effect produced with a fixed probability for observations when $A = 1$: measurement error ($P = 0.8$), selection on label ($P = 0.5$), and selection on predictors ($P = 0.8$).

On each dataset, we trained three predictors: a naive predictor trained on $A$ and $X$, a fairness through unawareness (FTU) predictor trained only on $X$, and a counterfactually fair predictor based on an average of the naive prediction under the assumption that $A = 1$ and the naive prediction under the assumption $A = 0$, weighted by the proportion of each group in the target distribution.

**Theorem 3.** *Let $X$ be an input dataset $X \in \mathcal{X}$ with a binary label $Y \in \mathcal{Y} = \{0, 1\}$ and protected class $A \in \{0, 1\}$. Define a predictor:*

$$f_{naive} := \underset{f}{\arg\min} \, \mathbb{E}[\ell(f(X, A), Y)]$$

*where $f$ is a proper scoring rule. Define another predictor:*

$$f_{CF} := \mathbb{P}(A = 1)f_{naive}(X, 1) + \mathbb{P}(A = 0)f_{naive}(X, 0)$$

*If the association between $Y$ and $A$ is purely spurious, then $f_{CF}$ is counterfactually fair.*

For robust prediction, we show that the counterfactually fair (CF) predictor has accuracy in the target distribution near the accuracy of a predictor trained directly on the target distribution. For group fairness, we show that the counterfactually fair predictor achieves demographic parity, equalized odds, and calibration, corresponding to the three biased datasets. Results are shown in Table 2 and Table 3, and code to reproduce these results or produce results with varied inputs (number of datasets sampled, effect of $A$ on $X$, probabilities of each bias, type of predictor) is available at https://github.com/jacyanthis/Causal-Context.

Table 3: Experimental results: Fairness metrics

| | Demographic Parity Difference (CF) | Equalized Odds Difference (CF) | Calibration Difference (CF) |
|---|---|---|---|
| Measurement Error | **-0.0005** | 0.0906 | -0.8158 |
| Selection on Label | 0.1321 | **-0.0021** | 0.2225 |
| Selection on Predictors | 0.1428 | 0.0789 | **0.0040** |

## 7 Discussion

In this paper, we provided a new argument for counterfactual fairness—that the supposed trade-off between fairness and accuracy [12] can evaporate under plausible conditions when the goal is accuracy in an unbiased target distribution (Theorem 1). To address the challenge of trade-offs between different group fairness metrics, such as their mutual incompatibility [30] and the variation in costs of certain errors such as false positives and false negatives [19], we provided a conceptual tool for adjudicating between them using knowledge of the underlying causal context of the social problem (Theorem 2 and Corollary 2.1). We illustrated this for the three most common fairness metrics, in which the bias of measurement error implies demographic parity; selection on label implies equalized odds; and selection on predictors implies calibration (Corollary 2.2), and we showed a minimal example by inducing particular biases on a simulated protected class in the Adult income dataset.

There are nonetheless important limitations that we hope can be addressed in future work. First, the counterfactual fairness paradigm still faces significant practical challenges, such as ambiguity and identification. Researchers can use causal discovery strategies developed in a fairness context [5, 20] to identify the causal structure of biases in real-world datasets and ensure theory like that outlined in this paper can be translated to application. Second, a key assumption in our paper and related work has been the assumption that associations between $Y$ and $A$ are "purely spurious," a term coined by Veitch et al. [47] to refer to cases where, if one conditions on $A$, then the only information about $Y$ in $X$ is in the component of $X$ that is not causally affected by $A$. This has provided a useful conceptual foothold to build theory, but it should be possible for future work to move beyond the purely spurious case, such as by articulating distribution shift and deriving observable signatures of counterfactual fairness in more complex settings [36, 39].

## 8 Acknowledgments

We thank Bryon Aragam, James Evans, and Sean Richardson for useful discussions.

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

# A Proofs

**Theorem 1.** *Let $\mathcal{F}^{CF}$ be the set of all counterfactually fair predictors. Let $\ell$ be a proper scoring rule (e.g., square error, cross entropy loss). Let the counterfactually fair predictor that minimizes risk on the training distribution $X, Y, A \sim P$ be:*

$$f^*(X) := \underset{f \in \mathcal{F}^{CF}}{\operatorname{argmin}} \, \mathbb{E}_P[\ell(f(X), Y)]$$

*Then, $f^*$ also minimizes risk on the target distribution $X, Y, A \sim Q$ with no selection effects, i.e.,*

$$f^*(X) = \underset{f}{\operatorname{argmin}} \, \mathbb{E}_Q[\ell(f(X), Y)]$$

*if either of the following conditions hold:*

1. *The association between $Y$ and $A$ is due to selection on label and the marginal distribution of the label $Y$ is the same in each distribution, i.e., $P(Y) = Q(Y)$.*

2. *The association between $Y$ and $A$ is due to selection on predictors.*

*Proof.* Counterfactual fairness is a case of counterfactual invariance. By Lemma 3.1 in Veitch et al. [47], this implies $X$ is $X_A^\perp$-measurable. Therefore,

$$\underset{f \in \mathcal{F}^{CF}}{\operatorname{argmin}} \, \mathbb{E}_P[\ell(f(X), Y)] = \underset{f \in \mathcal{F}^{CF}}{\operatorname{argmin}} \, \mathbb{E}_P[\ell(f(X_A^\perp), Y)]$$

Following the same reasoning as Theorem 4.2 in Veitch et al. [47], it is well-known that under squared error or cross entropy loss the risk minimizer is $f^*(x_A^\perp) = \mathbb{E}_P[Y \mid x_A^\perp]$. Because the target distribution $Q$ has no selection (and no confounding because $A$ is exogenous in the case of counterfactual fairness), the risk minimizer in the target distribution is the same as the counterfactually fair risk minimizer in the target distribution, i.e., $\mathbb{E}_Q[Y \mid x] = \mathbb{E}_Q[Y \mid x_A^\perp]$. Thus our task is to show $\mathbb{E}_P[Y \mid x_A^\perp] = \mathbb{E}_Q[Y \mid x_A^\perp]$.

Selection on label is shown in Figure 2b. Because $X_A^\perp$ does not d-separate $Y$ and $A$, $f^*(X)$ depends on the marginal distribution of $Y$, so we need an additional assumption that $P(Y) = Q(Y)$. We can use this with Bayes' theorem to show the equivalence of the conditional distributions,

$$Q(Y \mid X_A^\perp) = \frac{Q(X_A^\perp \mid Y)Q(Y)}{\int Q(X_A^\perp \mid Y)Q(Y)\mathrm{d}y} \tag{A.1}$$

$$= \frac{P(X_A^\perp \mid Y)Q(Y)}{\int P(X_A^\perp \mid Y)Q(Y)\mathrm{d}y} \tag{A.2}$$

$$= \frac{P(X_A^\perp \mid Y)P(Y)}{\int P(X_A^\perp \mid Y)P(Y)\mathrm{d}y} \tag{A.3}$$

$$= P(Y \mid X_A^\perp), \tag{A.4}$$

where the first and fourth lines follow from Bayes' theorem, the second line follows from the causal structure ($X$ causes $Y$), and the third line follows from the assumption that $P(Y) = Q(Y)$. This equality of distributions implies equality of expectations.

Selection on predictors is shown in Figure 2c. Because $X_A^\perp$ d-separates $Y$ and $A$, $f^*(X)$ does not depend on the marginal distribution of $Y$, so we immediately have an equality of conditional distributions, $Q(Y \mid X_A^\perp) = P(Y \mid X_A^\perp)$, and equal distributions have equal expectations, $\mathbb{E}_P[Y \mid x_A^\perp] = \mathbb{E}_Q[Y \mid x_A^\perp]$. $\square$

**Theorem 2.** *Let the causal structure be a causal DAG with $X_Y^\perp$, $X_A^\perp$, $Y$, and $A$, such as in Figure 2. Assume faithfulness between $A$, $Y$, and $f(X)$. Then:*

1. *Counterfactual fairness is equivalent to demographic parity if and only if there is no unblocked path between $X_A^\perp$ and $A$.*

2. *Counterfactual fairness is equivalent to equalized odds if and only if all paths between $X_A^\perp$ and $A$, if any, are either blocked by a variable other than $Y$ or unblocked and contain $Y$.*

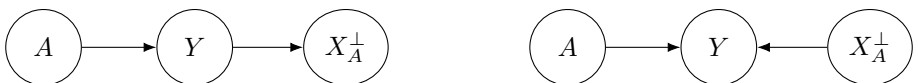

(a) Example of an unblocked path between $A$ and $X_A^\perp$    (b) Example of a blocked path between $A$ and $X_A^\perp$

Figure 3: Examples of an unblocked path and a blocked path in a causal DAG.

3. *Counterfactual fairness is equivalent to calibration if and only if all paths between $Y$ and $A$, if any, are either blocked by a variable other than $X_A^\perp$ or unblocked and contain $X_A^\perp$.*

*Proof.* For a predictor $f(X)$ to be counterfactually fair, it must only depend on $X_A^\perp$.

1. By Definition 1, demographic parity is achieved if and only if $X_A^\perp \perp A$. In a DAG, variables are dependent if and only if there is an unblocked path between them (i.e., no unconditioned **collider** in which two arrows point directly to the same variable). For example, Figure 3a shows an unblocked path between $A$ and $X_A^\perp$.

2. By Definition 2, equalized odds are achieved if and only if $X_A^\perp \perp A \mid Y$. If there is no path between $X_A^\perp$ and $A$, then $X_A^\perp$ and $A$ are independent under any conditions. If there is a blocked path between $X_A^\perp$ and $A$, and it has a block that is not $Y$, then $X_A^\perp$ and $A$ remain independent because a blocked path induces no dependence. If there is an unblocked path between $X_A^\perp$ and $A$ that contains $Y$, then $Y$ d-separates $X_A^\perp$ and $A$, so $X_A^\perp$ and $A$ remain independent when conditioning on $Y$. On the other hand, if there is a blocked path between $X_A^\perp$ and $A$ and its only block is $Y$, then conditioning on the block induces dependence. If there is an unblocked path that does not contain $Y$, then $X_A^\perp$ and $A$ are dependent.

3. With Definition 3, we can apply analogous reasoning to the case of equalized odds. Calibration is achieved if and only if $Y \perp A \mid X_A^\perp$. If there is no path between $Y$ and $A$, then $Y$ and $A$ are independent under any conditions. If there is a blocked path between $Y$ and $A$, and it has a block is not $X_A^\perp$, then $Y$ and $A$ remain independent because a blocked path induces no dependence. If there is an unblocked path between $Y$ and $A$ that contains $X_A^\perp$, then $X_A^\perp$ d-separates $Y$ and $A$, so $Y$ and $A$ remain independent when conditioning on $X_A^\perp$. On the other hand, if there is a blocked path between $Y$ and $A$ and its only block is $X_A^\perp$, then conditioning on the block induces dependence. If there is an unblocked path that does not contain $X_A^\perp$, then $Y$ and $A$ are dependent.

$\square$

**Corollary 2.1.** *Let the causal structure be a causal DAG with $X_Y^\perp$, $X_A^\perp$, $Y$, and $A$, such as in Figure 2. Assume faithfulness between $A$, $Y$, and $f(X)$. Then:*

1. *For a binary classifier, counterfactual fairness is equivalent to* **conditional demographic parity** *if and only if, when a set of legitimate factors $L$ is held constant at level $l$, there is no unblocked path between $X_A^\perp$ and $A$.*

2. *For a binary classifier, counterfactual fairness is equivalent to* **false positive error rate balance** *if and only if, for the subset of the population with negative label (i.e., $Y = 0$), there is no path between $X_A^\perp$ and $A$, a path blocked by a variable other than $Y$, or an unblocked path that contains $Y$.*

3. *For a binary classifier, counterfactual fairness is equivalent to* **false negative error rate balance** *if and only if, for the subset of the population with positive label (i.e., $Y = 1$), there is no path between $X_A^\perp$ and $A$, a path blocked by a variable other than $Y$, or an unblocked path that contains $Y$.*

4. *For a probabilistic classifier, counterfactual fairness is equivalent to* **balance for negative class** *if and only if, for the subset of the population with negative label (i.e., $Y = 0$), there is no path between $X_A^\perp$ and $A$, a path blocked by a variable other than $Y$, or an unblocked path that contains $Y$.*

5. *For a probabilistic classifier, counterfactual fairness is equivalent to* **balance for positive class** *if and only if, for the subset of the population with positive label (i.e., $Y = 1$), there is no path between $X_A^\perp$ and $A$, a path blocked by a variable other than $Y$, or an unblocked path that contains $Y$.*

6. *For a probabilistic classifier, counterfactual fairness is equivalent to* **predictive parity** *if and only if, for the subset of the population with positive label (i.e., $D = 1$), there is no path between $Y$ and $A$, a path blocked by a variable other than $X_A^\perp$, or an unblocked path that contains $X_A^\perp$.*

7. *For a probabilistic classifier, counterfactual fairness is equivalent to* **score calibration** *if and only if there is no path between $Y$ and $A$, a path blocked by a variable other than $X_A^\perp$, or an unblocked path that contains $X_A^\perp$.*

*Proof.* Each of these seven metrics can be stated as a conditional independence statement, as shown in Table 1, and each of the seven graphical tests of those statements can be derived from one of the three graphical tests in Theorem 2. Note that the graphical test for a binary classifier is the same as that for the corresponding probabilistic classifiers because the causal graph does not change when $f(X_A^\perp)$ changes from a binary-valued (i.e., $f(X) \in \{0, 1\}$) function to a probability-valued function (i.e., $f(X) \in [0, 1]$).

From demographic parity:

1. Conditional demographic parity is equivalent to demographic parity when some set of legitimate factors $L$ is held constant at some value $l$.

From equalized odds:

2. False positive error rate balance is equivalent to equalized odds when considering only the population with negative label (i.e., $Y = 0$).

3. False negative error rate balance is equivalent to equalized odds when considering only the population with positive label (i.e., $Y = 1$).

4. Balance for negative class is equivalent to equalized odds for probabilistic classifiers when considering only the population with negative label (i.e., $Y = 0$).

5. Balance for positive class is equivalent to equalized odds for probabilistic classifiers when considering only the population with negative label (i.e., $Y = 1$).

From binary calibration:

6. Predictive parity is equivalent to binary calibration when considering only the population with positive label (i.e., $D = 1$).

7. Score calibration is equivalent to binary calibration for probabilistic classifiers.

$\square$

**Corollary 2.2.** *Assume faithfulness between $A$, $Y$, and $f(X)$.*

1. *Under the graph with measurement error as shown in Figure 2a, a predictor achieves counterfactual fairness if and only if it achieves demographic parity.*

2. *Under the graph with selection on label as shown in Figure 2b, a predictor achieves counterfactual fairness if and only if it achieves equalized odds.*

3. *Under the graph with selection on predictors as shown in Figure 2c, a predictor achieves counterfactual fairness if and only if it achieves calibration.*

*Proof.* By Theorem 2:

1. Observe in Figure 2a that the only path between $X_A^\perp$ and $A$ is blocked by $Y$, so counterfactual fairness implies demographic parity. Because the only block in that path is $Y$, counterfactual fairness does not imply equalized odds. And the only path between $Y$ and $A$ (a parent-child relationship) is unblocked and does not contain $X_A^\perp$, so counterfactual fairness does not imply calibration.

2. Observe in Figure 2b that the only path between $X_A^\perp$ and $A$ is unblocked (because $S$ is necessarily included in the predictive model), so counterfactual fairness does not imply demographic parity. Because that path contains $Y$, counterfactual fairness implies equalized odds. And the only path between $Y$ and $A$ is unblocked (because $S$ is necessarily included in the predictive model) and does not contain $X_A^\perp$, so counterfactual fairness does not imply calibration.

3. Observe in Figure 2c that the only path between $X_A^\perp$ and $A$ is unblocked (because $S$ is necessarily included in the predictive model), so counterfactual fairness does not imply demographic parity. Because that path does not contain $Y$, counterfactual fairness does not imply equalized odds. And the only path between $Y$ and $A$ is unblocked (because $S$ is necessarily included in the predictive model) and contains $X_A^\perp$, so counterfactual fairness implies calibration.

$\square$

**Theorem 3.** *Let $X$ be an input dataset $X \in \mathcal{X}$ with a binary label $Y \in \mathcal{Y} = \{0,1\}$ and protected class $A \in \{0,1\}$. Define a predictor:*

$$f_{naive} := \underset{f}{\operatorname{argmin}} \, \mathbb{E}[\ell(f(X,A),Y)]$$

*where $f$ is a proper scoring rule. Define another predictor:*

$$f_{CF} := \mathbb{P}(A=1)f_{naive}(X,1) + \mathbb{P}(A=0)f_{naive}(X,0)$$

*If the association between $Y$ and $A$ is purely spurious, then $f_{CF}$ is counterfactually fair.*

*Proof.* Notice that $f_{CF}$ does not depend on $A$ directly because the realization of $A$ is not in the definition. To show that $f_{CF}$ also does not depend on $A$ indirectly (i.e., through $X$), consider that a purely spurious association means that $Y \perp X \mid X_A^\perp, A$. Therefore, the naive predictor:

$$f_{naive}(X,A) = \mathbb{P}(Y=1|X,A)$$
$$= \mathbb{P}(Y=1|X_A^\perp,A)$$

Because $X_A^\perp$ is the component of $X$ that is not causally affected by $A$, there is no term in $f_{CF}$ that depends on $A$, which means $f_{CF}$ is counterfactually fair. $\square$

