# OpenReview forum: "Causal Context Connects Counterfactual Fairness to Robust Prediction and Group Fairness"
_NeurIPS.cc/2023/Conference — NeurIPS 2023 poster_

### Official Review · Reviewer_c2Rt · 2023-07-04

**Soundness:** 2 fair
**Presentation:** 3 good
**Contribution:** 3 good
**Rating:** 6
**Confidence:** 3

**Summary:**

This work aims to provide guidance for contextually-informed algorithmic fairness criteria selection. The work motivates counterfactual fairness from its legal and ethical foundations. The primary contribution of the work is to detail a set of cases (data generating processes and selection mechanisms) where counterfactually fair predictors are accuracy-optimal in the (unselected) target distribution and consistent with specific correlational fairness criteria in the observed data.

**Strengths:**

*  The work is generally well-written and clear.
* The arguments and key results of this work provide a new perspective on cases where counterfactual fairness is consistent with predicting outcomes under particular selection mechanisms and data generating processes. While the results regarding the mapping between counterfactual fairness and observational fairness criteria largely follow from prior work on counterfactual invariance, this work does expand upon the prior work meaningfully.

**Weaknesses:**

Major:
* There are no experiments with either simulated or real-world data. At a minimum, it would be beneficial to include simulated experiments that verify the basic claims regarding the optimality of the counterfactually-fair predictors and the relationships between counterfactual fairness and the observational fairness criteria under different data generating processes.
* The restriction of the scope of the work to the purely spurious setting (line 201) limits the generalizability of the claims and this is largely not acknowledged. The work makes the claim that the purely spurious setting where association between group attributes and outcomes are due to selection is reasonable to cover the space of problems where fairness is of interest. I would argue that this is a serious limitation of the work because of the aim to provide guidance that broadly applies to the space of settings where it is of interest to evaluate fairness and bias considerations.
  *  For an example where the spurious setting may not hold, consider that in medical applications, structural racism and social determinants of health contribute to differences in observed health status across racial groups in addition to the differences in access to care that may be considered as selection effects. This can induce a difference in the conditional Y | X across subgroups (effectively an association between A and Y not mediated by X) that arises because race serves as a proxy for exposure to structural racism and social determinants. To the best of my understanding, this setting does not map cleanly on to the type of selection graphs considered in this work.

Minor:
* There is inconsistency in which variable is used to the ground truth outcome versus the noisy outcome with measurement error. In lines 112-113, Y-tilde is considered the noisy label and Y the ground truth. Elsewhere in the paper, Y-tilde is considered the ground truth.


**Questions:**

*  To what extent should the statements about the observable implications of counterfactual fairness be considered if and only if statements versus unidirectional statements? For instance, under Corollary 2.1.1 does demographic parity imply counterfactual fairness under the causal measurement error graph?
* Could you please add additional detail to prove the result regarding predictive parity? I generally follow the claim that binary calibration for binary classifiers is analogous to score calibration for risk scores, but it is not clear how conditioning on Y=1 yields predictive parity (which is confusingly referred to as D=1 in the supplement). It is also unclear whether the argument assumes that score calibration implies predictive parity. In general, this is not true (see Chouldechova 2017 Arxiv version, discussion immediately following definition 2.2 [1]).

1. Chouldechova, Alexandra. "Fair prediction with disparate impact: A study of bias in recidivism prediction instruments." arXiv preprint arXiv:1703.00056 (2017).

Minor suggestions:
*  Introduce the notion that a counterfactually fair predictor depends only on X_{A}^{\perp} in the main text rather than in the supplementary material
*  Replace “correlational” with “observational”
*  Replace “persecuted” with “prosecuted”


**Limitations:**

It would be beneficial for this work to expand upon the potential limits of the generalizability of its claims, especially due to the assumption of the purely spurious setting. Relatedly, it is not clear how well the claims are applicable outside of the set of graphs and selection mechanisms considered in this work.

---

> ### Author Rebuttal · Authors · 2023-08-10
>
> Thank you for the close read. We appreciate your time, and we believe that we have addressed your two major concerns by adding an experimental evaluation and clarifying the limitations (and purpose) of working in the purely spurious setting.
>
> > There are no experiments with either simulated or real-world data. At a minimum, it would be beneficial to include simulated experiments that verify the basic claims regarding the optimality of the counterfactually-fair predictors and the relationships between counterfactual fairness and the observational fairness criteria under different data generating processes.
>
> As suggested, we have added an experiment to verify the main claims of the paper. See the global response for details. We find that on a (semi-synthetic modification of) the Adult income dataset:
> 1. The counterfactually fair predictor trained on biased domain data has good performance when deployed on the unbiased target distribution (Theorem 1).
> 2. The counterfactually fair predictor satisfies the observational fairness criteria predicted by the underlying causal structure (Corollary 2.1).
>
> > The restriction of the scope of the work to the purely spurious setting (line 201) limits the generalizability of the claims and this is largely not acknowledged. The work makes the claim that the purely spurious setting where association between group attributes and outcomes are due to selection is reasonable to cover the space of problems where fairness is of interest. I would argue that this is a serious limitation of the work because of the aim to provide guidance that broadly applies to the space of settings where it is of interest to evaluate fairness and bias considerations.
>
> > For an example where the spurious setting may not hold ... this setting does not map cleanly on to the type of selection graphs considered in this work.
>
> > It would be beneficial for this work to expand upon the potential limits of the generalizability of its claims, especially due to the assumption of the purely spurious setting. Relatedly, it is not clear how well the claims are applicable outside of the set of graphs and selection mechanisms considered in this work.
>
> We see that this is an important limitation and appreciate the reviewer’s critique. We discuss this in the global response, but to make the purely spurious assumption more prominent in the paper, we have added it to the Introduction and to the Discussion (the final section). We have also elaborated on it on Page 6 to more fairly and thoroughly present the pros and cons when we formally introduce it.
>
> As discussed in the global response, assuming pure spuriousness substantially simplifies the results (and thus also the presentation), allowing us to thoroughly develop the connection from counterfactual fairness to out-of-domain robustness and observational fairness metrics. However, the purely spurious assumption is likely not necessary to make such a connection in general. We note that extending the results to more complicated causal scenarios and fairness definitions is an important direction for future work. (However, we think that the extra level of technical complexity would severely obfuscate the core point in the present paper).
>
> > There is inconsistency in which variable is used to the ground truth outcome versus the noisy outcome with measurement error. In lines 112-113, Y-tilde is considered the noisy label and Y the ground truth. Elsewhere in the paper, Y-tilde is considered the ground truth.
>
> We have made $\tilde{Y}$ the ground truth throughout.
>
> > To what extent should the statements about the observable implications of counterfactual fairness be considered if and only if statements versus unidirectional statements? For instance, under Corollary 2.1.1 does demographic parity imply counterfactual fairness under the causal measurement error graph?
>
> Observational data can strongly constrain causal structure but usually cannot uniquely identify it. Accordingly, we do not expect the results to be bidirectional in general. However, with the purely spurious assumption and faithfulness, the relationships seem bidirectional.
>
> > Could you please add additional detail to prove the result regarding predictive parity? ... It is also unclear whether the argument assumes that score calibration implies predictive parity…
>
> This is an error. Statement 2.2.6 should say “with positive outcome label (i.e., $D=1$)” instead of “with positive outcome (i.e., $Y=1$)”. We sincerely apologize for the mistake and the confusion. Please let us know if you would still like additional detail, but hopefully that clarifies the statement and proof.
>
> Regarding score calibration, if a score-calibrated predictor were converted into a binary predictor by using a score threshold, it would not necessarily imply predictive parity—as you mention is explained in Chouldechova (2017). The corrected version of Statement 2.2.6 does not mention score calibration, nor does its proof (lines 601-602), so we currently don’t believe the argument assumes this. Note that the proof only refers to binary calibration, which does not have the issue that Chouldechova raises in which $S | R = r$ (i.e., $S | A = a$) potentially differs across groups in ways that result in PPV imbalance.
>
> > Minor suggestions:
> > * Introduce the notion that a counterfactually fair predictor depends only on X_{A}^{\perp} in the main text rather than in the > supplementary material
> > * Replace “correlational” with “observational”
> > * Replace “persecuted” with “prosecuted”
>
> Done.

---

> ### Author Response · Authors · 2023-08-13
> **Follow-up with Reviewer c2Rt**
>
> Dear Reviewer c2Rt,
>
> Thank you again for the thorough review. We believe we have addressed each of your concerns, particularly by adding an experiment to validate the results and more thoroughly and prominently documenting the pros and cons of the purely spurious assumption. Could you please check our response and let us know if you have further questions?
>
> All the best,
>
> Authors

---

> > ### Comment · Reviewer_c2Rt · 2023-08-17
> >
> > Thank you for the thorough updates to the paper. I appreciate the effort that went into the additional experimental results and the response. My concerns have been addressed. I have updated my score from a 3 (reject) to a 6 (weak accept).

---

### Official Review · Reviewer_buKj · 2023-07-05

**Soundness:** 3 good
**Presentation:** 3 good
**Contribution:** 3 good
**Rating:** 7
**Confidence:** 4

**Summary:**

The authors show that under various causal graphs involving protected attributes, features and labels, that a counterfactually fair classifier achieves a specific association-based fairness metric. From this, they suggest a pipeline to detect counterfactual fairness.

**Strengths:**

The authors do a really good job making the point that counterfactual fairness, which is what is desired in a lot of theory, can be evaluated using more standard associational fairness metrics when assuming (or discovering) particular causal graphs.

The math seems to be correct.

The findings are useful to push causal fairness further into practice.

**Weaknesses:**

The organization of the paper can be improved. For example, a lot of the beginning of Section 4 would make more sense in the introduction. Figure 2 and its associated introductory text would make more sense in Section 5.

I don't think the impossibility theorems need to be emphasized so much. They're cute, but don't really give us anything practical, which this paper tries to do.

Figure 2: please put the corresponding correlational fairness metric in the main figure rather than only mentioning them in the caption.

Line 48: I don't think Ref [3] does data augmentation. However, this reference does: https://doi.org/10.1145/3375627.3375865.

Figure 1 mentions data generation and regularization, but they don't come up in the text. I think bringing up examples of each would be useful to the reader, e.g. https://proceedings.mlr.press/v161/ahuja21a.html and https://arxiv.org/abs/2002.10774.

It would be useful to discuss other theoretical work on the fairness-accuracy tradeoff, especially papers that relate this discussion to the presence or absence of different kinds of bias, e.g. https://proceedings.mlr.press/v119/dutta20a.html.

I understand the difficulty of empirical validation when dealing with counterfactuals, but the paper would be a lot stronger if the pipeline of Figure 1 were carried out to show the steps tangibly.

**Questions:**

Can you try to relate the evaporation of the tradeoff that you show to the argument made by Dutta et al. (reference given in the Weaknesses section of the review) and point out similarities and differences. It would also be good to relate the decomposition of X on line 268 to Dutta's other work https://doi.org/10.1609/aaai.v34i04.5794.

**Limitations:**

It would be good if the authors could be more explicit on how they obtain/discover the causal graph because it is quite important. That limitation hasn't been discussed enough. As part of that discussion, they may wish to mention causal fairness methods that do not require the entire causal graph a priori and are able to get pieces of it through a group-testing based approach which require a sublinear number of conditional independence tests.

---

> ### Author Rebuttal · Authors · 2023-08-10
>
> Thank you for the support and feedback on framing and content, which we believe has improved the paper.
>
> > The organization of the paper can be improved. For example, a lot of the beginning of Section 4 would make more sense in the introduction. Figure 2 and its associated introductory text would make more sense in Section 5.
>
> Per your suggestion, we have moved parts of Section 4 into the introduction. For the second move, we have moved discussion of the three scenarios (5.1, 5.2, 5.3) to the introduction in response to Reviewer U9Dm’s suggestion to, “Help your reader by making these connections right away,” which we think addresses your concern. We would also prefer Figure 2 to be early in the paper because it is so illustrative, if possible. We have also done some other reorganization as a result of other comments.
>
> > I don't think the impossibility theorems need to be emphasized so much. They're cute, but don't really give us anything practical, which this paper tries to do.
>
> The impossibility theorems motivate the need to make trade-offs between fairness metrics, but we probably did overemphasize them so we have now reduced the emphasis. We have also added a qualification here with reference to a 2023 FAccT paper that critiques their practical relevance [1] alongside the already-cited work that suggests a method to improve fairness across the different metrics “with minimal model performance reduction” [2]. We think this improves the motivation and contextualization of our results.
>
> [1] Bell, Bynum, Drushchak, Zakharchenko, Rosenblatt, and Stoyanovich. 2023. “The Possibility of Fairness: Revisiting the Impossibility Theorem in Practice.” FAccT. doi:10.1145/3593013.3594007.
>
> [2] Hsu, Mazumder, Nandy, and Basu. 2022. “Pushing the Limits of Fairness Impossibility: Who’s the Fairest of Them All?” NeurIPS. doi:10.48550/2208.12606.
>
> > Figure 2: please put the corresponding correlational fairness metric in the main figure rather than only mentioning them in the caption.
>
> Done.
>
> > Line 48: I don't think Ref [3] does data augmentation. However, this reference does: [link].
>
> Thank you for noticing this mistake. We have replaced it with your citation, Sharma et al. (2020).
>
> > Figure 1 mentions data generation and regularization, but they don't come up in the text. I think bringing up examples of each would be useful to the reader, e.g. [link] and [link].
>
> We have added brief mentions and citations to both Ahuja et al. (2021) and Di Stefano et al. (2020).
>
> > It would be useful to discuss other theoretical work on the fairness-accuracy tradeoff, especially papers that relate this discussion to the presence or absence of different kinds of bias, e.g. [link]. /// Can you try to relate the evaporation of the tradeoff that you show to the argument made by Dutta et al. (reference given in the Weaknesses section of the review) and point out similarities and differences.
>
> Thank you for the pointer; we’ve added a reference in the paper. The analysis of mismatched distributions by Dutta and colleagues provides an interesting complement to our own critique of the purported fairness-accuracy tradeoff. They base it in the different Chernoff information between groups, and we base ours on the performance of a counterfactually fair predictor in an unbiased target distribution. The two views are complementary is that one is grounded in information theory, the other in causal structure. We think that the results do not have a crisp mathematical connection in general.
>
> > I understand the difficulty of empirical validation when dealing with counterfactuals, but the paper would be a lot stronger if the pipeline of Figure 1 were carried out to show the steps tangibly.
>
> We have added an experiment with the Adult dataset illustrating the main claims of the paper. This stops short of illustrating all the steps (there’s limited space), but we think it helps clarify the empirical implications.
>
> > It would also be good to relate the decomposition of X on line 268 to Dutta's other work [link].
>
> Dutta and colleagues provide an information-theoretic decomposition that usefully separates discrimination into critical features that one wants to preserve and non-critical features that one is okay with being removed. Our decomposition is instead based in a causal framework, which separates the training data $X$ into the component not causally affected by $A$ and the component not causally affected by $Y$, respectively. We have added a brief discussion and citation to Dutta’s approach, which could be explored in future work.
>
> > It would be good if the authors could be more explicit on how they obtain/discover the causal graph … a group-testing based approach which require a sublinear number of conditional independence tests.
>
> Our technical results apply regardless of how the causal graph is obtained/discovered. Generally, we are envisioning that the analyst will use their understanding of the real-world situation under consideration to inform the causal graph they assume. We agree the need to (partially) articulate the causal graph is a limitation. It is a good point that the causal graph itself can be (partially) learned from data, and we have added a note that this is an interesting direction for future work, as well as a citation to Galhotra et al. (2022) [1], which develops a group-testing approach with sublinear conditional independence tests as you suggest.
>
> [1] Galhotra, Shanmugam, Sattigeri, and Varshney. 2022. “Causal Feature Selection for Algorithmic Fairness.” SIGMOD/PODS. doi:10.1145/3514221.3517909.

---

> ### Author Response · Authors · 2023-08-13
> **Follow-up with Reviewer buKj**
>
> Dear Reviewer buKj,
>
> We very much appreciate your constructive review and support. We believe that we have implemented each of your suggestions, which have in particular improved the contextualization of our results in the literature and, with the addition of experimental data, have provided some illustration of how our contribution can be applied. Could you please check our response and let us know if you have further questions?
>
> All the best,
>
> Authors

---

> > ### Comment · Reviewer_buKj · 2023-08-16
> >
> > The authors have done a ton of work, and have been very conscientious in responding to all the reviewers. I am satisfied that they have taken my suggestions to heart and worked on getting to a much better paper. I have raised my score.

---

### Official Review · Reviewer_7eza · 2023-07-06

**Soundness:** 4 excellent
**Presentation:** 4 excellent
**Contribution:** 4 excellent
**Rating:** 6
**Confidence:** 2

**Summary:**

This work investigated two important coutnterfactual fairness problems: (1) When will the counterfactually fair predictor also be the risk minimizer on the target distribution? (2) How do the classic statistical fairness metrics correspond to counterfactual fairness? In response to these questions, this work contextualized the problem in three causal DAG where selection bias may exist. To respond the first question, Theorem 1 shows under certain conditions that the counterfactual fair predictor is in fact accuracy-optimal. To respond the second question, Theorem 2 and Corollary 2.2 characterized the counterfactual fairness using the knowledge of the underlying causal context.  The success in responding the two questions will motivate a pipeline to achieve counterfactual fairness without performance degradation.



**Strengths:**

This work bridges the gap between counterfactual fairness and statistical fairness metrics in the presence of selection biases. The studied problem is novel, and the theoretical results are of high quality. The motivation and the presentation of the whole paper is clear, and can provide high-level insights for fairness researchers and practitioners.

**Weaknesses:**

I did not fully check the proof details but only read the sketch. It looks like the statements heavily depend on the three causal DAGs in Figure 2. For the theoretical results, the assumptions for each statement can be made more clearly.

**Questions:**

1. In Figure 2(b), $X^\perp_A$ is the effect of outcome $Y$. What is the implication of such causal structure in the real world?
2. In Figure 2(c), the selection variable $S$ is the causal effect of protected class $A$ and observed outcome $Y$. What if the selection $S$ is modeled as the cause of the outcome $Y$? Do the statements still hold?

**Limitations:**

 The authors did not address the limitations of how the counterfactual fairness can be evaluated in the real world.

---

> ### Author Rebuttal · Authors · 2023-08-10
>
> Thank you for the support, corrections, and provocative questions.
>
> > I did not fully check the proof details but only read the sketch. It looks like the statements heavily depend on the three causal DAGs in Figure 2. For the theoretical results, the assumptions for each statement can be made more clearly.
>
> Corollary 2.1 is exclusively about those three causal DAGs, but Theorem 1 and Theorem 2 neither reference nor depend on them. You make a good point that the scope of Corollary 2.1 is not very clear in the statement itself because Figure 2 is only mentioned on Line 245 in the preceding paragraph, so we have added a reference to Figure 2 in the statements to clarify that these statements refer to Figure 2a, Figure 2b, and Figure 2c, respectively.
>
> > Q1. In Figure 2(b), X \perp A is the effect of outcome Y. What is the implication of such causal structure in the real world?
>
> This causal structure is referred to as “anti-causal” and is the predominant setting in many machine learning applications. For example, in health care data, we’re often interested in assessing whether or not someone has a disease based on measurements of effects of that disease—e.g., $X$ may be a chest X-ray, and $Y$ some lung disease. Here, $Y$ causes $X$. (Ultimately, the true labels used for the training data may be produced by more expensive follow-up measurements or waiting for the disease to progress.)
>
> > Q2. In Figure 2(c), the selection variable S is the causal effect of protected class A and observed outcome Y. What if the selection S is modeled as the cause of the outcome Y? Do the statements still hold?
>
> We assume the reviewer means “and X” rather than “and observed outcome Y” because Figure 2(c) shows $S$ being affected by $X$, not $Y$. If so, note that $S$ is selection into the dataset. In our example, it is whether the loan repayment predictors $X$ and whether the person repaid ($Y$). Because the dataset contains historical data, it doesn’t seem to make logical sense in the real world to say that selection causes loan repayment.
>
> However, ignoring the real-world meaning of an effect of $S$ on $Y$, the mathematical answer is that if $S$ caused $Y$ in Figure 2(c), then counterfactual fairness would no longer imply calibration. This is because the new unblocked path between $A$ and $Y$ that does not go through $X_A^\perp$ makes the graph (i.e., the context) fail the test for calibration in Theorem 2. Additionally, in this case, $A$ would have a true causal influence on $Y$ (i.e., not purely spurious), and therefore it would be unclear whether a counterfactual notion of fairness is appropriate.
>
> > The authors did not address the limitations of how the counterfactual fairness can be evaluated in the real world.
>
> We describe a number of limitations of how counterfactual fairness can be evaluated in the real world, as detailed in Lines 150–182, beginning, “Counterfactual fairness is widely compelling, but there are still limitations.” We focus on two broad categories of limitation: the ambiguity of counterfactuals and the identification of causal structure.
>
> To make this more prominent, we have edited the discussion section to bring up these limitations. We would also note that some other revisions have also foregrounded the limitations of generalizability and application (e.g., the “purely spurious” assumption), and we have added an experiment with the Adult income dataset, in which we simulated a new protected class and induced the three causal contexts in Corollary 2.1 and showed that a counterfactually fair predictor achieves the corresponding metric and achieves the best performance in the unbiased target distribution (Theorem 1).

---

> ### Author Response · Authors · 2023-08-13
> **Follow-up with Reviewer 7eza**
>
> Dear Reviewer 7eza,
>
> We greatly appreciate your thoughtful review and support of the paper. We believe we have clarified the scope of when these statements hold and covered more real-world implications (including the addition of the Adult income experiment with a novel counterfactually fair predictor) and limitations that will help readers make use of our technical results. Could you please check our response and let us know if you have further questions?
>
> All the best,
>
> Authors

---

> > ### Comment · Reviewer_7eza · 2023-08-14
> > **My questions are addressed**
> >
> > I thank the authors for their insightful response. At this moment I do not have other questions. However, I would like to hold my score and further see other reviewers' discussions.

---

> > > ### Author Response · Authors · 2023-08-19
> > > **Thank you**
> > >
> > > Thank you for the kind words. As you said you wanted to wait for other reviewers, we just wanted to mention that there are only 48 hours left in the discussion period, and three others have now also responded to our rebuttals, so please let us know if any other questions have come up, and we will try to address them in time.

---

> > > > ### Comment · Reviewer_7eza · 2023-08-20
> > > > **Retain my score**
> > > >
> > > > All the reviewers agreed that the authors have adequately addressed their concerns. I would retain my score of 6.

---

### Official Review · Reviewer_U9Dm · 2023-07-08

**Soundness:** 3 good
**Presentation:** 3 good
**Contribution:** 3 good
**Rating:** 6
**Confidence:** 2

**Summary:**

The paper explores a connection between counterfactual fairness and correlation-based fairness metrics, exhibiting causal graph settings in which counterfactual fairness implies specific correlational metrics are satisfied. The paper further connects this mapping to the accuracy-fairness tradeoff.

**Strengths:**

I think the mapping between causal/counterfactual notions of fairness and metric-based fairness is useful. The paper does a good job of integrating several literatures into a coherent frame. The specific causal scenarios that relate counterfactual fairness and correlational fairness are useful, although I'm not well equipped enough in the structural causal modeling literature to understand the significance of this as a contribution.

**Weaknesses:**

I think the paper itself foresees my largest concern: it's not clear what settings these results apply in. While I'm not suggesting, as the conclusion tries to defend against, a full empirical validation of the theory, it would be nice if there was at least some attempt at mapping the causal graphs in Figure 2 to specific scenarios where the relationship being offered is relevant and an explanation of the relevance in that scenario. This exists, but way down in Section 5. Help your reader by making these connections right away.

The literature on fairness (the human value) as operationalized by fairness (statistical metrics) strongly suggests that the problem with choosing metrics is less a fairness/accuracy tradeoff or impossibility of satisfying multiple metrics, but the need to contextualize the model well in the application, understanding the cost and meaning of things like errors and the value of their avoidance. It's not obvious that connections to counterfactual fairness help with this problem, and if they don't, that's a limitation worth articulating.

The paper often cites into the literature in a way that only thinly and tenuously represents the large issues at play in the topic of the paper. I'd like to see more alignment between supporting sources and the content of the paper. For example, the citation of Jacobs & Wallach is offered as a problem with the difficulty of measurement error, but the paper is mostly about how to establish validity of models through explicit measurement modeling, a thing not considered by the methods in this paper at all.

Nit: the paper consistently uses the word "persecuted" instead of "prosecuted" (e.g., at 53, 264, and 266, and maybe elsewhere?)
Nit: the symbol l in Theorem 1 should probably be $\ell$.
Nit: the discussion of gender at 164 is highly oversimplified and is probably actually a description of biological sex. Later text clarifies this, but the claims are a little vague as currently stated.
Nit: "obtain" at 178 is probably the wrong word?

**Questions:**

How can the connections offered in this paper be oeprationalized for real-world cases? Many applications are mentioned, but the value of the connections is only loosely coupled to these.

**Limitations:**

There is some defensive posturing in the conclusion about the lack of real-world connection, which is probably a limitation to address by improving the connections rather than by disclaiming the possibility of doing this (since making those connections is the ostensible value of the paper).

---

> ### Author Rebuttal · Authors · 2023-08-10
>
> Thank you for your support and your useful comments.
>
> > I think the paper itself foresees my largest concern: it's not clear what settings these results apply in. … This exists, but way down in Section 5. Help your reader by making these connections right away.
>
> We agree that the mapping of causal graphs in Figure 2 to specific scenarios is important, and we have moved the example description scenarios to appear sooner in the paper.
>
> We have also added an experimental demonstration illustrating the results (see the global response), which we think further clarifies the practical scenarios corresponding to the causal graphs.
>
> > The literature on fairness (the human value) as operationalized by fairness (statistical metrics) strongly suggests that the problem with choosing metrics is less a fairness/accuracy tradeoff or impossibility of satisfying multiple metrics, but the need to contextualize the model well in the application, understanding the cost and meaning of things like errors and the value of their avoidance. It's not obvious that connections to counterfactual fairness help with this problem, and if they don't, that's a limitation worth articulating.
>
> Thank you for pointing out the importance of addressing the cost and meaning of errors that vary across contexts (e.g., false negatives versus false positives), and we have added that explicitly in the motivation. We believe notions of counterfactuals can potentially help with this problem. For example, the severity of harm from an error can be in part estimated by comparing it to the harm or benefit one would receive in the counterfactual scenario. And if one believes counterfactually unfair practices (i.e., those prohibited by the “but-for” test in U.S. law) lead to more harmful errors, then one might be able to use the correspondence we draw to better achieve that. For example, the types of errors made by a predictor that only achieves demographic parity will tend to be different than those of a predictor that only achieves equalized odds.
>
> We would push back a little on the problem not being a fairness/accuracy trade-off or the mutual impossibility. In our experience, the literature still views both of these as important challenges [e.g., 1 on the trade-off and 2 on impossibility, just as examples], and we see our work as contributing to the toolkit of methods to address them [e.g., 3 on improving fairness on multiple metrics with “minimal model performance reduction”]. We think that contextual factors, such as the causal graphs that are the focus of this work and other approaches to counterfactual fairness, are relevant for addressing all three of the issues: fairness/accuracy trade-off, the impossibility theorems, and the cost of errors and benefit of correct prediction/classification. We appreciate the reviewer helping us clarify this point, and we have added a brief discussion of this to the paper.
>
> [1] Ge, Zhao, Yu, Paul, Hu, Hsieh, and Zhang. 2022. “Toward Pareto Efficient Fairness-Utility Trade-off in Recommendation through Reinforcement Learning.” WSDM. doi:10.1145/3488560.3498487.
>
> [2] Hsu, Mazumder, Nandy, and Basu. 2022. “Pushing the Limits of Fairness Impossibility: Who’s the Fairest of Them All?” NeurIPS. doi:10.48550/arXiv.2208.12606.
>
> [3] Bell, Bynum, Drushchak, Zakharchenko, Rosenblatt, and Stoyanovich. 2023. “The Possibility of Fairness: Revisiting the Impossibility Theorem in Practice.” FAccT. doi:10.1145/3593013.3594007.
>
> > The paper often cites into the literature in a way that only thinly and tenuously represents the large issues at play … explicit measurement modeling, a thing not considered by the methods in this paper at all.
>
> We see that J&W are focused on a particular approach that we do not apply or respond to. As far as we know, they have the most rigorous development of “measurement error models” in the fairness context, which is why we cited them. We have edited line 262 to specify that we are referring to their development of such models, and we have also provided a more general citation for measurement error [1]. We also reviewed the other citations in the paper to ensure that they are aligned with our text.
>
> [1] Bland and Altman. 1996. “Statistics Notes: Measurement Error.” BMJ. doi:10.1136/bmj.312.7047.1654.
>
> > Nit: the paper consistently uses the word "persecuted" instead of "prosecuted" (e.g., at 53, 264, and 266, and maybe elsewhere?) Nit: the symbol l in Theorem 1 should probably be $\ell$.
>
> Fixed.
>
> > Nit: the discussion of gender at 164 is highly oversimplified and is probably actually a description of biological sex. Later text clarifies this, but the claims are a little vague as currently stated.
>
> We could move the later text to the beginning of the discussion and reword for clarity, but with the existing discussion of race, we have just cut the gamete randomization example to make room for suggested additions.
>
> > How can the connections offered in this paper be oeprationalized for real-world cases? Many applications are mentioned, but the value of the connections is only loosely coupled to these… /// There is some defensive posturing in the conclusion about the lack of real-world connection…
>
> We have provided a partial connection to real-world cases through:
> 1. clarification of how the connections help one to measure (or enforce) counterfactual fairness using observational data,
> 2. elaboration of the three example problems, and
> 3. an experiment demonstration of the main results (including a simple method for learning a counterfactually fair predictor in the purely spurious setting; see the global response).
>
> We have also expanded the discussion of what “purely spurious” means in practice and clarified that this assumption is a limitation of the present paper (though not a fundamental one—we think it’s a great direction for future work to extend the ideas here to circumvent this!). We also discuss this in more detail in the global response.

---

> ### Author Response · Authors · 2023-08-13
> **Follow-up with Reviewer U9Dm**
>
> Dear Reviewer U9Dm,
>
> Thank you for the support and clarifying comments. We believe we have addressed each of your concerns, including making the real-world connections more prominent and adding some empirical validation. Could you please check our response and let us know if you have further questions?
>
> All the best,
>
> Authors

---

> > ### Author Response · Authors · 2023-08-19
> > **48 hours left in discussion period**
> >
> > Dear Reviewer U9Dm,
> >
> > There are only 48 hours left in the discussion period. We greatly appreciate your review and are keen to know if the improvements we made in response have fully addressed your concerns. Thanks again.
> >
> > All the best,
> >
> > Authors

---

> > > ### Comment · Reviewer_U9Dm · 2023-08-22
> > >
> > > I apologize for chiming in late. Yes, these changes broadly address my concerns. I particularly appreciate the effort to improve the real-world scenarios and the separation of simplifying assumptions for presentation purposes from the complexities of real cases.
> > >
> > > On measurement modeling, there are a variety of useful general references from David Hand. My point was more that, in general, a lot of things are cited less because they are good authorities for the claim and more because they are tangentially related. For example, Jacobs &. Wallach are cited in a section "Causal graph with measurement error", but the issue in COMPAS (and all pretrial risk assessments) is more fundamental: re-arrest (the thing measured as a proxy) isn't just a noisy approximation of the variable of interest, but a thing with very different causal properties, so it's not even really clear that treating the gap as a noise term in the same graph is really making a good model vs needing to capture a different set of causal relationships to understand the nature of the sources of error. One could make a similar argument about the citation to Crenshaw, but on rereading that's more on target than I remembered.
> > >
> > > I don't know if the ACs want me to raise my score, but I'll move it up by one for avoidance of doubt.

---

> > > > ### Comment · Area_Chair_rGoq · 2023-08-22
> > > >
> > > > Please adjust your scores to reflect your latest view of the paper. Thank you!

---

### Official Review · Reviewer_WTYn · 2023-07-08

**Soundness:** 3 good
**Presentation:** 3 good
**Contribution:** 2 fair
**Rating:** 6
**Confidence:** 4

**Summary:**

This paper provides a new way to analyze the fairness-accuracy tradeoff in many fairness definition through the lens of counterfactual fairness. The authors prove that, under certain conditions, the counterfactually fair predictor is optimal when the target is OOD generalization. The theoretical analyses build the connection between counterfactual fairness and 10 well-known correlational fairness.


**Strengths:**

S1. Interesting way to connect counterfactual fairness, out-of-distribution generalization and other fairness definitions.

S2. Technically sound proof on the theorems and corollaries.

S3. Writing quality is good.

**Weaknesses:**

W1. Some statements/assumptions need more comprehensive illustration.

W2. Interesting theoretical analyses on counterfactual fairness

W3. The authors use impossibility theorem as one motivation, but more discussion about the technical analyses to impossibility theorem would better position this paper.

W4. Title is a bit misleading to me.

**Questions:**

Q1. I agree that assumptions about the purely spurious relationship between $Y$ and $A$ helps with theorem 1. But for the race and zip code example in lines 203 -- 205, can we simply say the bias is due to selection effect? It is obvious that population from different demographic groups are distributed unevenly in a community, and the selection effect may further amplify the bias. Can we simply characterize it as due to selection effect only?

Q2. Is it possible to elaborate more on the assumption about decomposing $X$ into $X_{\perp A}$ and $X_{\perp Y}$? Is it a linear decomposition of $X$? It is a pretty strong assumption to decompose $X$ and I am not sure whether it is reasonable or not.

Q3. What is the relationship between the proposed counterfacutal fairness-based analysis to impossibility theorem? The authors use impossibility theorem as one motivation for studying counterfactual fairness, but no more discussion about how these analyses are related to impossibility theorem is included.

Q4. Where is the evaluation on simulated data? The authors mention it as a contribution but I could not find it. If I miss it, it would be good to provide pointers.

Q5. How does the proposed analyses help **select** the algorithmic fairness metrics? Some discussions (or maybe a case study) about how to use the proposed analyses to select fairness metrics would help.

**Limitations:**

L1. Many theoretical analysis in the paper assume the decompostion of $X$ into two parts. But it is unclear whether this assumption is reasonable.

L2. It might be good to draw figures for some concepts for better clarity (e.g., illustrative figures for unblocked path and the correspondence between counterfactual fairness and correlational fairness). And it would be good to recall Figure 2 when discussing related technical details (e.g., theorem 2).

L3. The authors mention that the theorems hold on simulated data, but no evaluation is found.

L4. It would be great to better connect the content with the title `select algorithmic fairness metrics'.

L5. Please carefully check typos in the paper, e.g., `if they had if they were classified negatively' $\rightarrow$ `if they were classfied negatively'.

================== Post-rebuttal ==================
The authors have addressed my concern so I am raising my score.

---

> ### Author Rebuttal · Authors · 2023-08-10
>
> Thank you for your time and useful suggestions, which significantly improve the manuscript.
>
> > W1. Some statements/assumptions need more comprehensive illustration.
>
> We have added the more comprehensive illustrations in response to your comments below.
>
> > W3/Q3. The authors use impossibility theorem as one motivation, but more discussion about the technical analyses to impossibility theorem would better position this paper…
>
> We only intended  the impossibility theorems to illustrate the need to make context-specific fairness decisions in practice because the metrics cannot all be satisfied at once, except in the unrealistic cases of perfect prediction, random prediction, or equal features across groups [1,2]. The technical counterfactual fairness analysis relates to this impossibility by providing a tool to select between the observational metrics based on the situation’s causal context (i.e., the data-generating process). We state the criteria that the causal context has to meet in Theorem 2 and Corollary 2.2. In response to the reviewer’s helpful suggestion, we have added a brief explanation of this relationship to the Discussion. We have also reduced the emphasis on impossibility theorems as a motivation in the Abstract and Introduction because there are many other reasons why we need to select between fairness metrics and the impossibility results may not be as relevant in practice as it may seem [e.g., 3].
>
> [1] Chouldechova. 2017. “Fair Prediction with Disparate Impact: A Study of Bias in Recidivism Prediction Instruments.” Big Data. doi:10.1089/big.2016.0047.
>
> [2] Kleinberg, Mullainathan, and Raghavan. 2016. “Inherent Trade-Offs in the Fair Determination of Risk Scores.” ITCS. doi:10.48550/arXiv.1609.05807.
>
> [3] Andrew, Bynum, Drushchak, Zakharchenko, Rosenblatt, and Stoyanovich. 2023. “The Possibility of Fairness: Revisiting the Impossibility Theorem in Practice.” FAccT. doi:10.1145/3593013.3594007.
>
> > W4/L4. Title is a bit misleading to me … better connect the content with the title `select algorithmic fairness metrics'.
>
> We plan to retitle the paper “Causal Context Connects Counterfactual Fairness to Group Fairness and Robust Prediction.”
>
> > Q1. I agree that assumptions about the purely spurious relationship between Y and A helps with theorem 1. But for the race and zip code example in lines 203 -- 205, can we simply say the bias is due to selection effect? It is obvious that population from different demographic groups are distributed unevenly in a community, and the selection effect may further amplify the bias. Can we simply characterize it as due to selection effect only?
>
> We appreciate you pointing out that correlations between race and recidivism might be due to other factors in addition to the selection effect. We have softened the language here and clarified our statement that “other factors” can have effects because we agree that, in the real world, it is probably not only due to a selection effect.
>
> > Q2/L1. Is it possible to elaborate more on the assumption about decomposing X into X \perp A and X \perp Y? Is it a linear decomposition of X?...
>
> We have added more detail when we first mention decomposition. We do not mean a necessarily linear composition. The decomposition is quite general, and does not specify a particular functional form. E.g., $X^\perp_A$ may be some arbitrarily complicated nonlinear function $g(X)$, so long as $g(X(A=0)) = g(X(A=1))$ a.s. A key advantage of the view in this paper is that we do not need to know what this function is explicitly. The decomposition into two (abstract) parts is the ‘purely spurious’ assumption—see the global response for a discussion. We have added text to clarify this in the paper.
>
> > Q4/L4. Where is the evaluation on simulated data? The authors mention it as a contribution but I could not find it…
>
> We apologize for the erroneous statement left over from a draft that had simulations. We have in fact now added semi-synthetic experiments based on the Adult income dataset, detailed in the global response.
>
> > Q5. How does the proposed analyses help {\bf select} the algorithmic fairness metrics? Some discussions (or maybe a case study) about how to use the proposed analyses to select fairness metrics would help.
>
> In short: the analyst uses their understanding of the real-world situation to posit a causal graph, then selects the fairness metric implied by that graph. We give examples in the paper of measurement error in arrest data (e.g., COMPAS), selection on outcome with people of younger age or privileged backgrounds being more likely to be included in the dataset, and selection on predictors with loan repayment data also not being a balanced mix of all protected classes. We discuss real-world application more in the global response.
>
> > L2. It might be good to draw figures for some concepts … recall Figure 2 when discussing related technical details (e.g., theorem 2).
>
> We have added a brief illustration of step-by-step reasoning to connect counterfactual fairness to observational fairness. (e.g., an example of an unblocked path). We have also added an explicit mention of Figure 2 to the statement of Theorem 2.
>
> > L5. Please carefully check typos in the paper, e.g., ```if they had if they were classified negatively' $\rightarrow$``` if they were classfied negatively'.
>
> We have fixed this and gone through the paper to check for other typos.

---

> ### Author Response · Authors · 2023-08-13
> **Follow-up with Reviewer WTYn**
>
> Dear Reviewer WTYn,
>
> We are grateful for your suggestions and comments. We believe we have addressed each of your concerns and made each of the improvements you suggest, such as adding an experiment on the Adult income dataset and clarifying the decomposition. Could you please check our response and let us know if you have further questions?
>
> All the best,
>
> Authors

---

> > ### Comment · Reviewer_WTYn · 2023-08-17
> > **Response to author rebuttal**
> >
> > Dear authors,
> >
> > I appreciate your efforts in addressing my concerns. I don't have further concerns and will raise my score.
> >
> > Best,
> > Reviewer WTYn

---

### Author Rebuttal · Authors · 2023-08-10

We thank the reviewers for their time and thorough commentary. We are glad to hear the core technical results were seen as valuable contributions (“interesting way to connect…”, “technically sound”, “useful”, “novel”, “the theoretical results are of high quality”, “expand upon the prior work meaningfully”) and the general presentation was clear (“integrating several literatures into a coherent frame”, “clear”, “really good job making the point”, “generally well-written and clear”).

There were two main reviewer concerns: experimental evaluation of the theoretical results and real-world applicability.

# Experiments
As several reviewers suggested, we conducted an experimental test of the main theoretical predictions of the paper. See the attached PDF for the table of results. Per NeurIPS policy, we have sent a code link to the AC privately.

As predicted, we find:
1. A counterfactually fair predictor satisfies the observational fairness metric corresponding to the underlying causal graph (Corollary 2.1).
2. Counterfactually fair predictors trained on biased data (where the protected class is associated with the outcome) have strong predictive performance on the unbiased target domain where no association exists (Theorem 1).

To conduct this experiment,
1. We produce a novel counterfactually fair (CF) predictor for the purely spurious setting, $f_{CF}(X) = P(Y=1|X, A=1)P(A=1) + P(Y=1|X, A=0)P(A=0)$ (i.e., a weighted average of naive predictors across each level of the protected class). This follows because $P(Y=1| X, A) = P(Y=1|X_A^\perp,A)$ in the spurious setting. Then $f_{CF}$ only depends on $X_A^\perp$, so it is in fact CF. $f_{CF}$ can be readily estimated in practice. Namely, fit a model $f_{naive}(X,A)$ that predicts $Y$ from both $X$ and $A$ (thus estimating $P(Y=1 | X, A)$) and then let $f_{CF}(X) = f_{naive}(X, 1) P(A=1) + f_{naive}(X, 0) P(A=0)$.
2. We use the Adult income dataset to construct semi-synthetic data matching each of the causal graphs. We first construct a synthetic binary protected class $A$, sampled at random. We then have $A$ causally affect $X$ by fixing the value of several attributes with a fixed probability when $A=1$ (if $A=0$, no change is made). We then create three biased datasets corresponding to the three causal graphs shown in Figure 2 by (a) noising $Y$ according to $A$, (b) selecting data points based on $A$ and $Y$, and (c) selecting on $A$ and predictors of $Y$.
3. We train CF predictors on each biased dataset then check the performance of the CF predictor on the base data where $A$ and $Y$ are independent (Theorem 1) and check observational fairness metrics (Corollary 2.1).

# Real-world connections and the ‘purely spurious’ assumption
The other concern that reviewers raised is about real-world applicability. In particular, there is a concern that the ‘purely spurious’ requirement is too strong in practice and is not sufficiently acknowledged as a key limitation.

We basically agree with this point! We view the contribution of the paper as novel technical results that show counterfactual fairness is both measurable and desirable in some natural situations. The significance of this, which the reviewers appreciated, is:
1. Counterfactual fairness is an important notion socially (e.g., it maps to a legal notion of discrimination).
2. It’s not obvious a priori that counterfactual fairness can ever be assessed or connected to other observational fairness metrics. We show that considering causal structure can allow this.
3. Prominent existing work [1] suggests that counterfactual fairness results in an unacceptable penalty to performance. Our results on out-of-domain performance directly rebut this by showing counterfactual fairness can be desirable even if one is concerned only with performance.
Focusing on the purely spurious setting lets us make these points clearly, avoiding obfuscation from extra technical complexity. We follow the recent literature in using this assumption [e.g., 2,3,4] to make meaningful progress in this challenging area.

We also think it should be possible for future work to move beyond the purely spurious case. There is a literature on causal fairness beyond counterfactual fairness that offers guidance on how to define fairness notions in more complex scenarios [e.g., 5, 6]. Given an assumed causal structure and a desired causal fairness notion respecting this structure, one should be able to extend these results to articulate the kind of domain shifts where a fair predictor is expected to be robust and to derive observable signatures of the fairness notion. We do not believe we have room in a single paper to thoroughly develop the core technical results and also tackle these complexities, but this is an exciting direction for future work.

As suggested by the reviewers, we have also changed the text to foreground the purely spurious assumption as a limitation. In response to comments that the paper would be stronger with real-world application (e.g., “if the pipeline of Figure 1 were carried out to show the steps tangibly”), we have conducted the experimental test but also foregrounded the role that we see these results playing in that broader pipeline.

[1] Nilforoshan, Gaebler, Shroff, and Goel. 2022. “Causal Conceptions of Fairness and Their Consequences.” PMLR. doi:10.48550/2207.05302.

[2] Makar and D’Amour. 2023. “Fairness and Robustness in Anti-Causal Prediction.” TMLR. doi:10.48550/2209.09423.

[3] Makar, Packer, Moldovan, Blalock, Halpern, and D’Amour. 2022. “Causally Motivated Shortcut Removal Using Auxiliary Labels.” AIStats. doi:10.48550/2105.06422.

[4] Veitch, D’Amour, Yadlowsky, and Eisenstein. 2021. “Counterfactual Invariance to Spurious Correlations in Text Classification.” NeurIPS. doi:10.48550/2106.00545.

[5] Plecko and Bareinboim. 2022. “Causal Fairness Analysis.” doi:10.48550/2207.11385.

[6] Nabi, Razieh, and Ilya Shpitser. 2018. “Fair Inference On Outcomes.” AAAI. doi:10.1609/aaai.v32i1.11553.

---

### Decision · Program_Chairs · 2023-09-21

**Decision:**

Accept (poster)

**Comment:**

The paper connects counterfactual fairness with correlational fairness and explores its relationship with out-of-distribution generalization. The approach is interesting, technically sound, and the paper is well-written. There were some questions and concerns raised by reviewers including the connection to impossibility theorems, the assumption of decomposing variables, and the connection between title and paper's actual content. The paper would benefit from addressing these comments.